# Cross-Sectional Study on Relationships Among FoMO, Social Influence, Positive Outcome Expectancy, Refusal Self-Efficacy and SNS Usage

**DOI:** 10.3390/ijerph17165907

**Published:** 2020-08-14

**Authors:** Kun-Hua Lee, Chia-Yu Lin, Jing Tsao, Lien Fang Hsieh

**Affiliations:** Department of Educational Psychology and Counselling, National Tsing Hua University, Hsin-chu City 300, Taiwan; lovejesusyuu@gmail.com (C.-Y.L.); angelia861106@gmail.com (J.T.); hsieh_1107@gapp.nthu.edu.tw (L.F.H.)

**Keywords:** FoMO, social cognitive theory, social influence, self-efficacy, positive outcome expectancy

## Abstract

Objectives: Use of social networking sites (SNS; i.e., Facebook or Instagram) is common, and people use SNS to communicate and share information. Literature indicates the extent of SNS usage could be influenced by fear of missing out (FoMO). FoMO means a process of appraisal and psychological need for SNS use. This study proposes a model that integrates three determinants of social cognitive theory (SCT) to explain the impact of FoMO on SNS usage. Design: A cross-sectional study was conducted using data from 259 participants recruited from a website. Main Outcome Measures: The analysis focused on FoMO, social influence, positive outcome expectancy, refusal self-efficacy, and SNS-related behavior cloud-based sites. Data are examined using descriptive analysis and structural equation modeling. Results: The proposed model reported proper goodness of fit. FoMO did not directly or indirectly impact SNS usage through the determinants of SCT. However, social influence and refusal self-efficacy had a direct effect. Conclusions: The roles of the three determinants of SCT vary by stage of SNS usage. FoMO and refusal self-efficacy are more strongly related with SNS addiction. Further research, particularly longitudinal and intervention studies, is needed to examine the effects of specific factors on SNS addiction.

## 1. Introduction

The use of social networking sites (SNS), such as Instagram and Facebook, to communicate and share information has become commonplace. Studies in European countries show that 70% of adolescents use SNS, and among them, 40% spend two hours or more on these sites on a daily basis [1]. The literature also highlights the negative effects of SNS, despite its high accessibility and convenience [2]. For example, a survey on mental health among college students revealed that 23.5% of the students could not resist using SNS, and a higher percentage used the sites for more than four hours per day. Further, these students reported higher levels of depression and symptoms of anxiety [3], compulsive use [4], and addiction [5]. Therefore, a psychosocial mechanism for SNS addiction is necessary.

Recently, fear of missing out (FoMO) has become a critical indicator of SNS usage and its negative effects [6]. FoMO is defined as “a pervasive apprehension that others might be having rewarding experiences from which one is absent” and is “characterized by a desire to stay continually connected with others” [7,8,9]. FoMO can be treated as a type of motivation or intention to connect with others. Self-determinant theory (SDT) has been applied as a theoretical basis for FoMO and SNS usage [6]. The theory posits that different types of motivations and psychological needs facilitate an individual’s desire to chase or do something such as healthy behaviors or social activities [10].

FoMO exists in social environments, and while it is rooted in SDT [11], the theory is developed through laboratory experiments, casting doubt on its ability to sufficiently explain the psychosocial mechanism of FoMO on SNS usage [10]. In addition, the definition of FoMO differs from those proposed in the past. The current definition accounts for psychological needs or a process of appraisal, for example, evaluating if information is present or absent [12]. This highlights the need for a comprehensive model for FoMO and SNS usage rooted in a social environment [13] cross-sectional study that investigates the effects of FoMO, psychological distress, and attitude toward SNS usage on users (n = 511) and shows that the cognitive factors of SNS usage influence SNS usage and addiction. The relationship among FoMO, SNS-specific cognition, and SNS usage warrants further clarification.

Social cognitive theory (SCT) has been widely applied to explain unhealthy behaviors (e.g., addiction) [14,15,16]. Bandura, the founder of social cognitive theory, argued human behaviors are influenced by cognitive vicarious, self-reflective, and self-regulatory processes [17]. Moreover, people could display or change their behavior through observation and cognitive appraisal. The cores of SCT assume several important factors, including refusal self-efficacy and outcome expectancy, and determine human behaviors [18] and social norms [19]. Related to SNS use, SCT believes humans could be diffused by acquiring and adopting information in a social network [20]. Further, the theory suggests that individuals who report greater positive outcome expectancies and lower self-efficacy to resist social influences are more likely to exhibit addictive behaviors (Figure 1) [21].

Refusal self-efficacy is defined as an individual’s beliefs that he or she can resist or refuse something (e.g., drugs or web browsing) [22]. A cross-sectional study [23], for example, assessed university students’ refusal self-efficacy in terms of mobile phone usage and outcome expectancies. Their results found a significant impact by refusal self-efficacy and outcome expectancy on mobile phone usage and that having close relationships did not affect the level of refusal self-efficacy. Literature supports the effect of refusal self-efficacy on internet use behavior and SNS use [24]. Therefore, we further examined the relationship among refusal self-efficacy and other factors related to SNS usage behavior.

Outcome expectancy is an individual’s beliefs about consequences that follow successful performance [25]. Outcome expectancy is a significant predictor of addictive behaviors (i.e., SNS addiction) [26]. The findings of [27] a cross-sectional study support the role of outcome expectancy as a mediator between negative emotional status and SNS addiction. Their study further indicated that depressive individuals with stronger feelings toward peer support tended to spend more time on SNS.

Social influence is broadly defined as the effects of social norms on individuals’ efforts to gain a more accurate understanding to effectively respond to social situations [28]. More specifically, the attitude of others could pressure an individual into making decisions consistent with social norms [29]. Prior study [30] found that social influence impacted the intention to use SNS through compliance with peer identification.

Several studies have adopted SCT to explain SNS addiction [22,31]. The study [24], for example, examined SNS addiction, attitudes toward SNS usage, and refusal self-efficacy among a youth population and found that refusal self-efficacy and outcome expectancies particularly impacted SNS addiction. However, few analyses have been conducted on the relationships among FoMO, SCT determinants, and SNS usage. Thus, this study proposes and applies an SCT-based model to examine FoMO and SNS usage.

The literature defines FoMO as a feeling of inadequacy, irritability, and anxiety and the impulsive checking of information online and on SNS [32]. Individuals with greater FoMO are more sensitive to social requests and support embedded in their online social networks [33]. Studies also show that individuals with greater positive outcome expectancy cannot resist SNS usage [22]. All constructs and definitions of the hypothesized model are shown in Table 1.

The present study aims to propose a more comprehensive psychosocial model to explain the relationship between FoMO and SNS. We assumed FoMO could have an indirect effect on SNS use through different kinds of social cognitive determinants, including stronger peer pressure to use SNS or lower self-efficacy to refuse using SNS, even if they could have more positive outcome expectancies for using SNS. The findings can be applied to develop more effective assessment instruments and interventions. Figure 2 depicts the proposed SCT-based model for the relationship between FoMO and SNS usage. This study tests the following two hypotheses.

**Hypothesis** **1.**
*FoMO indirectly affects SNS usage through social influence.*


**Hypothesis** **2.**
*Drawing on SCT, social influence indirectly affects SNS usage through positive outcome expectancy or refusal self-efficacy.*


## 2. Methods

### 2.1. Procedures and Participants

This study was approved by the National Tsing Hua University’s Institutional Review Board (IRB No. 10805HT041). Community population was the main resource of sampling, and our participants were recruited as they could read and write and habitually used SNS. There were no exclusive sampling criteria. The total number of participants was 268. The analysis focused on four age groups on the basis of the distribution of national representation: 20–29 years (N = 83) (mean age =22.87 years, S.D. = 2.331 years), 30–39 years (N = 79) (mean age = 33.32 years, S.D. = 3.217 years), 40–49 years (N = 50) (mean age = 44.56 years, S.D. = 3.131 years), and 50–65 years (N = 46) (mean age = 55.93 years, S.D. = 4.187 years). Only one questionnaire did not provide age. The research assistant posted the link to questionnaires on the website. The participants could connect to the link if they were willing to participate in this study. Participants who provided their informed consent were asked to fill out questionnaires shared via Google Docs. After excluding incomplete questionnaires, the number of eligible questionnaires analyzed was 259 (mean age = 36.10 years and S.D. = 12.40 years; average internet usage time = 141.06 min/day and S.D. = 108.97 min). There was no significant age difference between incomplete and complete questionnaires (F=2.38, *p* = 0.124). The online questionnaires were based on SNS-related behaviors, the frequency and degree of FoMO, positive outcome expectancy of internet usage, and refusal self-efficacy. Participants who completed all of the questionnaire were paid NT $100.

### 2.2. Measurements

#### 2.2.1. SNS Usage

Participants were asked if they used Facebook, Facebook Messenger, Line, Instagram, and other SNS platforms. They were awarded one point for using each SNS platform. The highest scores of SNS use was five, and the lowest score was zero. The number of “yes” responses and the average time of daily SNS usage (in minutes) were then estimated.

#### 2.2.2. FoMO Scale

The scale used 10 items to assess FoMO. The items were rated using a five-point Likert scale, where 1 denotes “completely disagree” and 5 is “completely agree.” Higher scores indicated greater levels of FoMO [32]. This study adopted the three FoMO subscales of relatedness, autonomy, and competence from Mandarin version FoMO scale [33]. The internal reliabilities of these scales in the context of this study were as follows: 0.84 (total score), 0.78 (relatedness), 0.78 (autonomy), and 0.47 (competence).

#### 2.2.3. Positive Outcome Expectancy

A positive outcome expectancy scale was developed to assess the level of positive outcome expectancy for internet usage [22]. All items were rated on a five-point Likert scale. Higher scores denote a greater degree of positive outcome expectancies. A factor analysis conducted using varimax revealed six factors of positive outcome expectancy: relax and boost confidence, gain knowledge, make friends, have fun, experience autonomy, and convenience. However, convenience was excluded owing to its lower factor loading. Cronbach’s α for all other factors was 0.92, 0.91, 0.92, 0.87, and 0.81, respectively.

#### 2.2.4. Refusal Self-Efficacy

A refusal self-efficacy questionnaire with 35 items measured on a six-point Likert scale. A higher total score denotes strong self-efficacy to refuse internet usage [34]. A factor analysis performed using varimax highlights six factors of refusal self-efficacy: leisure and kill time, search for information and interpersonal, work, relaxation, routine activity, and exploration for novelty. Cronbach’s α for each factor was 0.95, 0.83, 0.71, 0.76, 0.79, and 0.88, respectively.

#### 2.2.5. Social Influence

Three items were used to measure the degrees of social influence: peers’ positive attitude toward SNS usage, family’s positive attitude toward SNS usage, and SNS accessibility. A five-point Likert scale was used to assess the items. A higher score indicated greater social influence. Cronbach’s α for social influence is 0.75.

### 2.3. Statistical Analyses

This study employed several goodness-of-fit indices to assess the fitness of the model: χ^2^ test, root-mean-square error of approximation (RMSEA < 1.0), standardized RMR (SRMR < 0.8), comparative fit index (CFI < 1.0), and goodness-of-fit index (GFI < 1.0) [34]. In addition, structural equation modeling (SEM) was used to examine the fitness of the model. Finally, a bootstrap analysis was conducted to explore the mediating effects. Significance was set to 0.05.

## 3. Results

### 3.1. Description of Measured Variables

Table 2 shows the distributions of main categories in the present study. Table 3 presents the distributions and correlations for the measured variables. Among the 20–29 years group, averaged time of SNS use was 166.51 min (S.D. = 120.471 min); 91.6% of participants reported using Facebook, 89.2 % of them used Instagram, 97.6% of them reported using Line, and 80.7% of them reported using Messenger. Among the 30–39 years group, averaged time of SNS use was 153.92 min (S.D. = 105.52 min); 88.6% of participants reported using Facebook, 62% of them used Instagram, 93.7 % of them reported using Line, and 62% of them reported using Messenger. Among the 40-49 years group, the average time of SNS use was 128.20 min (S.D. = 111.345 min); 90% of participants reported using Facebook, 28% of them used Instagram, 96% of them reported using Line, and 46% of them reported using Messenger; Among the 50–65 years group, the average time of SNS use was 83.59 min (S.D.=56.849 min); 78.3% of participants reported using Facebook, 17.4% of them used Instagram, 91.3% of them reported using Line, and 37% of them reported using Messenger. There was a significant difference of time for SNS use among the four age groups (F = 6.834, *p* = 0.000). The results of Fisher’s Least Significant Difference post hoc test showed significant differences between the 20-29 group and the 40–49 years group (*p* = 0.043) and between the 20–29 years group and the 50–65 years group (*p* = 0.000). In other words, SNS usage decreased as age increased. As for the distribution of SNS usage, 88% of the participants used Facebook (n = 228), 56.4% used Instagram (n = 146), 95% used Line (n = 246), 60.6% used Facebook Messenger (n = 157), and 4.6 % used other SNS platforms (n = 12).

Further, there was no significant difference between age groups and the use of Facebook (χ^2^ = 5.34, *p* = 0.15) and Line (χ^2^ = 2.87, *p* = 0.41). However, there were significant differences between the use of Instagram (χ^2^ = 82.01, *p* = 0.00) and Facebook Messenger (χ^2^ = 29.34, *p* = 0.00) and various age groups. The post-hoc test results showed that a higher proportion of participants aged younger than 40 used Instagram (47.7%) and Facebook Messenger (45%), whereas these rates were lower for participants in the older age groups (Instagram = 8.5%, Facebook Messenger = 15.5%).

### 3.2. Goodness-Of-Fit Indices

The SEM results indicated that the proposed model employed proper goodness-of-fit indices (χ^2^ = 2.98, *p* = 0.00; GFI = 0.86; RMSEA = 0.09; SRMR = 0.13). Social influence significantly impacted positive outcome expectancy (t = 3.58, *p* < 0.001) and affected SNS usage (t = 2.55, *p* = 0.01). Refusal self-efficacy showed a direct effect on SNS usage (t = −3.87, *p* < 0.001). An unexpected finding is that several paths did not reach significant levels: FoMO–social influence (t = −0.27, *p* = 0.79), social influence–refusal efficacy (t = 0.45, *p* = 0.65), FoMO–SNS (t = −1.75, *p* = 0.08), and positive outcome expectancy–SNS (t = 1.81, *p* = 0.07).

## 4. Discussion

This study adopts social cognitive theory to explain the relationship between FoMO and SNS usage. Despite the proposed model reporting a good fit, the results failed to support the mediated effect of social influence between FoMO and SNS usage. Further, our findings do not support the impact of FoMO on SNS usage. Nevertheless, the study confirms the direct influence of social influence and refusal self-efficacy on SNS usage.

First, this study hypothesizes that social influence plays a mediating role between FoMO and SNS usage, although the results do not support this hypothesis. It then hypothesizes that SNS usage results from social influence, including compliance and conformity; therefore, individuals with greater FoMO spend more time using SNS to maintain relationships [28]. Contrary to our hypothesis, individuals with higher FoMO spend more time on SNS, and this behavior is independent of conformity or compliance. A possible explanation is that FoMO could be an outcome and not a cause of SNS usage. The literature suggests that FoMO is a specific type of internet addiction [12]. Further, users with greater FoMO spend more time on SNS owing to their lower self-efficacy to resist the temptation of SNS rather than their need to conform to social norms. We re-examined the proposed model for FoMO, refusal self-efficacy, and SNS usage and found satisfactory goodness-of-fit indices (χ^2^ = 2.58, *p* = 0.00; GFI = 0.93; CFI = 0.91; RMSEA = 0.08; SRMR = 0.07). The paths of FoMO–refusal self-efficacy, FoMO–SNS usage, and refusal self-efficacy–SNS usage reached significant levels (t = −2.4, *p* = 0.02; t = 2.55, *p* = 0.01; t = −4.12, *p* < 0.001). Further experimentation or longitudinal research is needed to examine the effects of refusal self-efficacy and FoMO on SNS usage.

Second, social influence significantly impacts SNS usage despite it failing to mediate the relationship between FoMO and SNS usage. Our results are supported by existing studies [35]. Similar to the development of drug use, social influence is key in initiating rather than maintaining SNS usage [36]. Therefore, it is important to develop an effective preventive program for SNS usage with a focus on proper social skills and self-assertive training. Further, research should focus on the various roles of risk factors during different stages of SNS addiction and, accordingly, design specific and effective intervention programs.

Finally, the findings of this study do not support the effects of outcome expectancy on SNS usage. Research suggests that positive outcome expectancy indirectly impacts refusal self-efficacy [22]. In other words, individuals with stronger refusal self-efficacy may decrease their SNS usage despite their greater positive outcome expectancies. Further research is needed to develop strategies of refusal self-efficacy in the context of SNS usage or addiction [37].

## 5. Conclusions

This study proposes a model for SNS usage and FoMO on the basis of three determinants of social cognitive theory. While our results fail to confirm the effects of FoMO and the mediated role of social influence, they indicate that social influence and refusal self-efficacy have varying roles during the different stages of SNS usage. In particular, refusal self-efficacy and FoMO are highly related with SNS addiction.

This study is subject to several limitations that warrant consideration. First, the cross-sectional design could not confirm the causality of FoMO and SNS usage. Thus, researchers should consider a longitudinal or intervention study. Second, our study does not assess the effects of negative emotional status on FoMO and SNS usage. The literature suggests that negative emotional status could be a risk factor in SNS usage [9,31]. In addition, we did not collect more demographic data, as the biases of some sample characteristics (i.e., gender or years of education) could not be excluded. Therefore, demographic information still needs to be considered in the future. Finally, we examine social influence using three self-reported items despite them failing to show proper reliability. Future studies could adopt more reliable measurements to assess social influence.

Our results can be used as a basis to develop effective intervention programs for SNS usage and a framework to assess the psychosocial conceptualization of SNS users and addiction through clinical practices.

## Figures and Tables

**Figure 1 ijerph-17-05907-f001:**
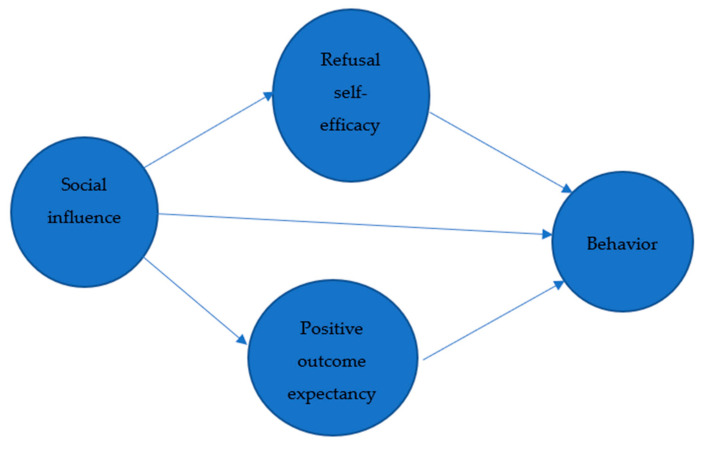
The theoretical model of social cognitive theory. FoMO: fear of missing out; SNS use: Social networking sites use. References 14,15,16 are missing. Please revise references in numerical order

**Figure 2 ijerph-17-05907-f002:**
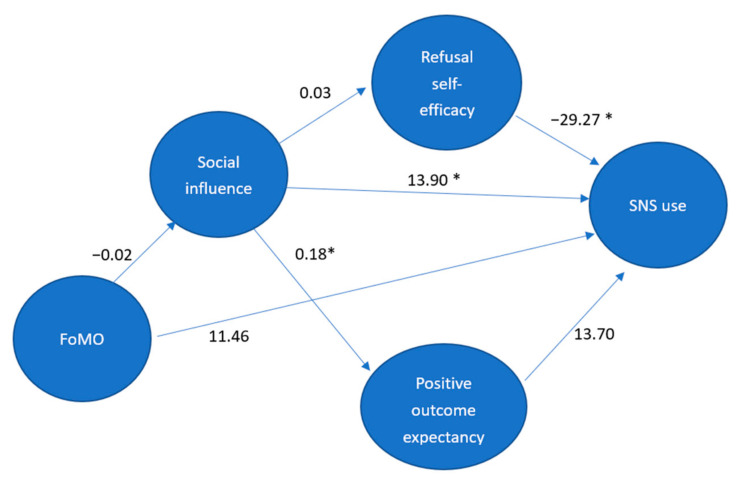
Estimates in hypothesized model. FoMO: fear of missing out; SNS use: Social networking sites use; *: *p* < 0.05.

**Table 1 ijerph-17-05907-t001:** The constructs and definitions of the hypothesized model.

Construct	Definition
Fear of Missing Out	The current definition accounts for psychological needs or a process of appraisal, for example, evaluating if information is present or absent
Social influence	Defined as the effects of social norms on individuals’ efforts to gain a more accurate understanding to effectively respond to social situations
Refusal self-efficacy	Defined as individuals’ beliefs that they can resist or refuse something (e.g., drugs or web browsing)
Positive outcome expectancy	Defined as individuals’ beliefs about consequences that follow successful performance
SNS use	A kind of social networking site (SNS), such as Instagram and Facebook, to communicate and share information through the use of different wireless devices.

**Table 2 ijerph-17-05907-t002:** The distributions of main categories.

Main Category	Mean/Standard Deviation	Range (Min–Max)
Fear of Missing Out	2.5/0.747	1.00–5.00
Refusal self-efficacy	3.07/0.777	1.00–5.00
Positive outcome expectancy	3.15/0.664	1.20–5.00
Social influence	4.16/0.690	1.33–5.00
SNS use	3.05/1.030	0.00–6.00

**Table 3 ijerph-17-05907-t003:** The correlation and distributions of measured variables.

	1	2	3	4	5	6	7	8	9	10	11	12	13	14	15	16	17	18	19
1	-	0.20 **	0.20 **	0.03	0.29 **	0.17 **	0.02	0.22 **	0.16 **	0.13 *	0.12	0.18 **	0.09	−0.27 **	−0.15 *	−0.17 **	−0.14 *	−0.01	−0.21 **
2		-	0.11	0.01	0.19 **	0.13 *	0.14 *	0.12	0.19 **	0.13 *	0.02	0.14 *	0.14 *	−0.22 **	−0.11	−0.07	−0.17 **	−0.05	−0.24 **
3			-	0.67 **	0.56 **	0.02	−0.10	−0.02	0.27 **	0.26 **	0.17 **	0.07	0.15 *	0.02	0.05	0.07	0.09	0.05	−0.01
4				-	0.30 **	−0.08	−0.15 *	−0.15 *	0.13 *	0.17 **	0.13 *	0.02	0.13 *	0.22 **	0.20 **	0.13 *	0.08	0.06	0.09
5					-	0.01	−0.16 *	−0.01	0.14 *	0.24 **	0.12 *	0.04	0.06	−0.1	0.02	−0.03	0.05	0.03	−0.07
6						-	0.57 **	0.58 **	0.49 **	0.29 **	0.34 **	0.55 **	0.36 **	−0.10	−0.10	0.04	−0.13 *	−0.09	−0.13 *
7							-	0.46 **	0.36 **	0.07	0.19 **	0.47 **	0.27 **	−0.01	−0.09	−0.01	−0.03	−0.16 *	−0.07
8								-	0.47 **	0.26 **	0.32 **	0.50 **	0.33 **	−0.16 *	−0.18 **	−0.14 *	−0.26 **	−0.16 *	−0.26 **
9									-	0.51 **	0.50 **	0.70 **	0.59 **	−0.08	−0.09	−0.01	−0.12	−0.03	−0.11
10										-	0.45 **	0.41 **	0.50 **	−0.06	−0.00	−0.05	−0.03	−0.01	−0.06
11											-	0.44 **	0.39 **	0.09	0.06	0.04	−0.09	−0.10	0.03
12												-	0.50 **	−0.01	−0.03	0.02	−0.17 **	−0.09	−0.12
13													-	0.00	0.07	−0.00	−0.02	−0.02	−0.08
14														-	0.60 **	0.56 **	0.50 **	0.07	0.55 **
15															-	0.44 **	0.33 **	0.01	0.45 **
16																-	0.23 **	0.06	0.39 **
17																	-	0.26 **	0.50 **
18																		-	0.18 **
19																			-
M	3.05	141.06	4.11	3.83	4.54	2.61	1.94	2.92	3.00	4.02	2.85	2.28	3.12	2.95	3.13	3.18	3.45	3.36	3.05
S.D.	1.03	108.97	0.84	0.96	0.71	0.90	0.92	0.85	0.86	0.71	1.08	0.84	0.97	0.95	1.07	1.08	1.02	1.32	1.09

*: *p* < 0.05; **: *p* < 0.001; 1: sum of categories for Social Networking Sites use; 2: averaged times of SNS use; 3: peer’s positive attitudes toward using SNS; 4: family’s positive attitudes toward using SNS; 5: accessibility for SNS use; 6: relatedness subscale of Fear of Missing Out; 7: autonomy subscale of FoMO; 8: competency subscale of FoMO; 9: relax and enhancement of confidence subscale of positive outcome expectancy; 10: gain knowledge subscale of positive outcome expectancy; 11: make friends subscale of positive outcome expectancy; 12: having fun subscale of positive outcome expectancy; 13: autonomy subscale of positive outcome expectancy; 14: leisure and kill time subscale of refusal self-efficacy; 15: information seeking and interpersonal subscale of refusal self-efficacy; 16: work time subscale of refusal self-efficacy; 17: relaxation subscale of refusal self-efficacy; 18: routine activity subscale of refusal self-efficacy;19: novelty seeking subscale of refusal self-efficacy; M: mean; S.D.: standard deviation.

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
