# Peer review of "Cross-Sectional Study on Relationships Among FoMO, Social Influence, Positive Outcome Expectancy, Refusal Self-Efficacy and SNS Usage"

_ijerph, 2020, doi:10.3390/ijerph17165907_

Round 1

Reviewer 1 Report

How exactly participants were recruited is not described; please describe the way in which subjects were recruited, and whether this could have introduced selection bias. Apart from age other participant characteristics are not described, yet this may have significant effects on the data interpretations and conclusions. Please provide more detail about the subject population.

The authors call attention to an increasingly important problem, given the large amount otime youth spend on social media and the effects on less salubrious aspects of behavior. They posit an appropriate theroetical schema to test, with good use of hte prior literature. Seslf-determinant theory as they poit out has been studied via experiments, and differeing definitions require a ocmprehensive model.

The study is good in the numbers of subjects and broad age distribution. Their interpretations and conclusions so far are reasonable, but as they point out various factors such as negative emotional status and ongoing measurements are needed to further understand these phenomena.

Author Response

Comment 1. How exactly participants were recruited is not described; please describe the way in which subjects were recruited.

Answer 1. Thanks for your comment. Our participants were recruited from website. They could connect the link of questionnaire if they are willing to participate this study. They completed questionnaires online after informed consents. Therefore, exactly sample size was 268, including incomplete and complete questionnaires (please to see p.3.).

Comment 2. Whether this could have introduced selection bias.

Answer 2. Thanks for your comment. We think selective bias could be reduced as recruiting and complete questionnaire online in this study.

Comment 3. Apart from age other participant characteristics are not described, yet this may have significant effects on the data interpretations and conclusions. Please provide more detail about the subject population.

Answer 3. Thanks for your comment. After review literature ((Tomczyk & Selmanagic-Lizde, 2018).) and ethical issue, we found demographic data could have lower effect on FoMO and SNS use. Therefore, this is a limitation of the present study (please to see p.8).

Comment 4. The authors call attention to an increasingly important problem, given the large amount of time youth spend on social media and the effects on less salubrious aspects of behavior. They posit an appropriate theoretical schema to test, with good use of the prior literature. Self-determinant theory as they point out has been studied via experiments, and different definitions require a comprehensive model.

Answer 4. Thanks for your comment. We did want to provide alternative psychological framework to explain the relationships among cognitive determinants, FoMO and SNS use.

Comment 5. The study is good in the numbers of subjects and broad age distribution. Their interpretations and conclusions so far are reasonable, but as they point out various factors such as negative emotional status and ongoing measurements are needed to further understand these phenomena.

Answer 5. Thanks for your suggestions. We have added your suggestions in the section of conclusion (please to see 2nd limitation in p. 8).

Reviewer 2 Report

Thank you for the opportunity to review this paper.  Authors sought to report findings from their study seeking to test a model that integrates determinants of cognitive theory to explain the impact of fear of missing out on social networking usage.  The paper would be stronger with the following revisions.  Introduction-Here the stage is set on the health issue and model.  I recommend a table that has the constructs and definitions.  I found myself going back and forth finding the constructs and definitions to understand the model.  I recommend complimentary text along with the table to complete thoughts and connecting the ideas.  I thought the Refusal Self Efficacy construct was not as fully developed for this model.  I presume, based on the example, that any non-compliance in a health regimen would be viewed as refusal.  I take issue with this.  Very much like Roger's Diffusion of Innovation, there are stages to adoption and this construct is addressing this idea of a laggard. Methods-The sample is representative of various age groups and that is good.  What is unclear is the method of recruitment, inclusion and exclusion criteria.  How many surveys were sent and not returned?  Was there missing data and how were these addressed? Who developed the survey and what is its history in research, prior administration and reliability and validity? Discussion-Third sentence "...our findings do not prove..." recommend support rather than prove.  Use of prove here is an exaggerated term-this study wouldn't prove anything but it can support.  I don't know the year of this study and question the impact of COVID on findings.  I wonder if social media usage has increased for information seeking as a portion of fear of missing out-missing out with information.  Thank you.

Author Response

Thank you for the opportunity to review this paper.  Authors sought to report findings from their study seeking to test a model that integrates determinants of cognitive theory to explain the impact of fear of missing out on social networking usage.  The paper would be stronger with the following revisions. 

Introduction-Here the stage is set on the health issue and model. 

Comment 1. I recommend a table that has the constructs and definitions. I found myself going back and forth finding the constructs and definitions to understand the model.  I recommend complimentary text along with the table to complete thoughts and connecting the ideas. 

Answer 1. Thanks for your suggestion. We not only provided a new table for these constructs, but also added some illustrations in p.3.

Comment 2. I thought the Refusal Self Efficacy construct was not as fully developed for this model.  I presume, based on the example, that any non-compliance in a health regimen would be viewed as refusal.  I take issue with this. Very much like Roger's Diffusion of Innovation, there are stages to adoption and this construct is addressing this idea of a laggard.

Answer 2. Thanks for your comments. In our study, refusal self-efficacy could be treated as the extent of self-efficacy to refuse using SNS when they encounter some high risk situations, for example, stressful life or frustration or on. Therefore, non-compliance may not fully reflect the exact extent of people confidently reject to use SNS. Sometimes, people who are non-compliance still display some behaviors despite they don’t want to do. Besides, literature also used Refusal self-efficacy to examine the extent of Internet addiction (For example, Wu, A. M., Cheung, V. I., Ku, L., & Hung, E. P. (2013). Psychological risk factors of addiction to social networking sites among Chinese smartphone users. Journal of behavioral addictions2(3), 160-166.).

Methods-The sample is representative of various age groups and that is good. 

Comment 3.What is unclear is the method of recruitment, inclusion and exclusion criteria. 

Answer 3. Thanks for your comment. After review literature ((Tomczyk & Selmanagic-Lizde, 2018).) and ethical issue, we found demographic data could have lower effect on FoMO and SNS use. Therefore, we enrolled the participants based on age. We alos added a limitation of the present study in conclusion (please to see p.8).

Comment 4. How many surveys were sent and not returned? 

Answer 4. Thanks for your comments. Based on our data in the Cloud, there were 268 questionnaires in the present study. Nine questionnaires were incomplete data.

Comment 5. Was there missing data and how were these addressed?

Answer 5. Thanks for your comments. Among nine incomplete questionnaires, the participants did not completely answer all questionnaires. In order to provide more precise information, the data without duration of surfing web were treated as missing data.

Comment 6. Who developed the survey and what is its history in research, prior administration and reliability and validity?

Answer 6. Thanks for your comments. Our research team designed and uploaded questionnaires in Cloud, and we posted the link of questionnaires in the website. 

Comment 7. Discussion-Third sentence "...our findings do not prove..." recommend support rather than prove. 

Answer 7. Thanks for your suggestion, we have revised it.

Comment 8. Use of prove here is an exaggerated term-this study wouldn't prove anything but it can support. 

Answer 8. Thanks for your suggestion, we will notice the difference between prove and support.

Comment 9. I don't know the year of this study and question the impact of COVID on findings.  I wonder if social media usage has increased for information seeking as a portion of fear of missing out-missing out with information. 

Answer 9. Thanks for your comments, it could be happened after pandemic. Maybe further study could examine the impact of COVID on SNS use.

Reviewer 3 Report

Thank you for the opportunity to review this article. The article use structural equation modeling to study the relationships among FoMO, social cognitive components and SNS use. The topic is very interesting and the authors did a lot of analyses. I hope the following are useful considerations:

  1. Page 2, paragraph2 : "SCT argues...". The sentence may be misleading since it has multiple "and".               
  2. Introduction, SCT theory:SCT has several different aspects(e.g. social reinforcement, social expectation, observational learning, behavior capacity...), I hope the authors can discuss SCT as a whole and then discuss which items of the theory specifically apply to this topic. The readers may have a better understanding of this area with sufficient SCT background knowledge.                                                             
  3. Introduciton: To my knowledge, social norm is not in SCT theory. Please make a clear distinction between SCT and social norm when refering to the two concepts in the introduction.                                                                                                                                 
  4. Overall, the introduction part present SCT theory that support the hypotheses in the manuscript. I hope the authors can better organize the theory part in the introduction so that readers not familiar with the topic can have a better understanding.                                        
  5. Figure 1, Behavior is misspelled.                                                        
  6. It would be great if the author can include both latent variable and measured variables in figure 2. I know the author listed out measurements in method section but it is not very straightforward that what variables are used to represent latent variables.                                                       
  7. For figure 2, I wonder whether reciprocal determinism plays a role in the process.                                                                                         
  8. For 2.2.1 SNS usage number of "yes", did the author use any of the SNS platform usage as "yes"? Or Sum numbers of "yes" with the usage of each platform? I hope the authors can describe the questions clearly.                                                                             
  9. For 2.2.2 FoMO scale, can the author put the 10 items in appendix?    
  10. For 2.2.4 Refusal self-efficacy, can the authors put the 35 items in the appendix or briefly describe what components are? Just want to have a basic understanding of the refusal self-efficacy questionnaire.         

Author Response

  1. Page 2, paragraph2 : "SCT argues..."  The sentence may be misleading since it has multiple “.

Answer: Thanks for your comments, we have revised it.              

  1. Introduction, SCT theory: SCT has several different aspects (e.g. social reinforcement, social expectation, observational learning, behavior capacity...), I hope the authors can discuss SCT as a whole and then discuss which items of the theory specifically apply to this topic. The readers may have a better understanding of this area with sufficient SCT background knowledge. 

Answer: Thanks for your comments. We have added some illustrations in page

  1. Introduction: To my knowledge, social norm is not in SCT theory. Please make a clear distinction between SCT and social norm when referring to the two concepts in the introduction. 

Answer: Thanks for your comment. Social norm is the last component of SCT. SCT focus on the effect of normative conduct on human behaviors, for example, people will imitate peer’s behaviors in order to follow their social norm. Therefore, this study still treated social norm as a component of SCT.  

  1. Overall, the introduction part present SCT theory that support the hypotheses in the manuscript. I hope the authors can better organize the theory part in the introduction so that readers not familiar with the topic can have a better understanding. 

Answer: thanks for your comment, we have added some illustrations in page 2. 

  1. Figure 1, Behavior is misspelled.   

Answer: thanks for your comment, we have revised it.

  1. It would be great if the author can include both latent variable and measured variables in figure 2. I know the author listed out measurements in method section but it is not very straightforward that what variables are used to represent latent variables. 

Answer: Thanks for your comment. The goal of the present study was to examine the hypothesized model of determinants of SCT and FoMO on SNS use. Therefore, we wanted to focus on the index of goodness-of-fit. Besides, there could be massy if add measurements in figure 2. Due to two reasons mentioned above, we still did not add measurements in Figure 2.  

  1. For figure 2, I wonder whether reciprocal determinism plays a role in the process. 

Answer: Thanks for your comment. We thinks reciprocal determinism could be examined in a longitudinal study. However, cross-sectional study could not provide the results of reciprocal determinants. Further study could examine the causal relationships among these factors.

  1. For 2.2.1 SNS usage number of "yes", did the author use any of the SNS platform usage as "yes"? Or Sum numbers of "yes" with the usage of each platform? I hope the authors can describe the questions clearly.   

Answer: Thanks for your comments. SNS usage was counted by summing all of SNS use platform. We have added some illustrations in 2.2.1.

  1. For 2.2.2 FoMO scale, can the author put the 10 items in appendix? 

Answer: Thanks for your comments. Considering copyright of FoMO scale, we could not put 10 items in appendix. If you want to use FoMO scale, you can mail to Dr. Przybylski for permission.

  1. For 2.2.4 Refusal self-efficacy, can the authors put the 35 items in the appendix or briefly describe what components are? Just want to have a basic understanding of the refusal self-efficacy questionnaire. 

Answer: Thanks for your comments. Considering copyright of refusal self-efficacy scale, we could not put these items in appendix. If you want to use FoMO scale, you can mail to Dr. Lin for permission. Besides, we can provide related references for you.

Reviewer 4 Report

Lee et al., performed a cross-sectional study to analyze the relationship between FoMO (Fear of Missing Out) and SNS (Social Networking Sites) by using a SCT-based model. Three psychological determinants of behavior of SCT model were used: positive outcome expectancy, refusal self-efficacy and social influence.

The topic of research is interesting, the manuscript is well written and the used methods are appropriated. However, there are some results that can be better presented, some aspects of manuscript need to be clarified and some incongruences present in the text should be revised. These are addressed below section by section:

UNDER ABSTRACT:

  • Please define what FoMO and SNS mean.

UNDER METHODS:

  • The authors state that the number of participants were 268 but the eligible ones were 259. Why were 9 people excluded from the study? In general, the authors should also clarify in the text if they applied any exclusion criteria (for example people with some neuropsychiatric disorder or some condition that not fit with the purpose of the study).
  • Under “Procedures and participants” paragraph, the authors state that the total number of participants were 259 but sum of subjects included in the four age groups is 260 (80+74+64+42). Please, carefully revise the text and correct the incongruence.

UNDER RESULTS:

  • Under “description of measured variables” paragraph, the authors should represent in a table these results as they did for the Table 1. They should at least include the four age groups with the mean age, the ratio Male/Female and % of internet usage (each SNS platform should be include).
  • “Convenience” subscale of “positive outcome expectancy” determinant of SCT model is lacking in table 1. Please add the missing results in the table.
  • For a faster interpretation of results, the authors should include a main row on the top part of table 1 with the name of main categories analyzed (SNS usage, social influence, Fomo Scale, positive outcome expectancy and refuse self-efficacy) and the score range and/or measurement unit used for each category or subscale.

MINOR CONCERNS:

  • Some typographical errors are present in the text and figures. Please carefully revise the whole manuscript and correct them.

Author Response

Lee et al., performed a cross-sectional study to analyze the relationship between FoMO (Fear of Missing Out) and SNS (Social Networking Sites) by using a SCT-based model. Three psychological determinants of behavior of SCT model were used: positive outcome expectancy, refusal self-efficacy and social influence.

The topic of research is interesting, the manuscript is well written and the used methods are appropriated. However, there are some results that can be better presented, some aspects of manuscript need to be clarified and some incongruences present in the text should be revised. These are addressed below section by section:

UNDER ABSTRACT:

  • Please define what FoMO and SNS mean.

Answer: Thanks for your comments, we have added some illustrations in abstract.

UNDER METHODS:

  • The authors state that the number of participants were 268 but the eligible ones were 259. Why were 9 people excluded from the study? In general, the authors should also clarify in the text if they applied any exclusion criteria (for example people with some neuropsychiatric disorder or some condition that not fit with the purpose of the study).

Answer: Thanks for your comments, we have some illustrations in sampling.  

  • Under “Procedures and participants”paragraph, the authors state that the total number of participants were 259 but sum of subjects included in the four age groups is 260 (80+74+64+42). Please, carefully revise the text and correct the incongruence.

Answer: Thanks for your comments. We have revised it.

UNDER RESULTS:

  • Under “description of measured variables” paragraph, the authors should represent in a table these results as they did for the Table 1. They should at least include the four age groups with the mean age, the ratio Male/Female and % of internet usage (each SNS platform should be include).

Answer: Thanks for your comments. We have added some illustrations in p. 5-6.

  • “Convenience” subscale of “positive outcome expectancy” determinant of SCT model is lacking in table 1. Please add the missing results in the table.

Answer: Thanks for your comment. Due to lower factor loading of subscale of convenience, the present study did not examine the relationship among convenience and other variables.  

  • For a faster interpretation of results, the authors should include a main row on the top part of table 1 with the name of main categories analyzed (SNS usage, social influence, Fomo Scale, positive outcome expectancy and refuse self-efficacy) and the score range and/or measurement unit used for each category or subscale.

Answer: Thanks for your comments, we have a Table 2 to show the distributions of main categories.

MINOR CONCERNS:

  • Some typographical errors are present in the text and figures. Please carefully revise the whole manuscript and correct them.

Answer: Thanks for your comment. We have checked and revised the typos as possible as we can.

Round 2

Reviewer 2 Report

Thank you for the opportunity to review this revised version of the paper.  My concerns have been addressed and have strengthened the paper.  Thank you.

Reviewer 3 Report

Thank you for the opportunity to review the paper again! My comments are addressed and the revised version looks good to me.

Reviewer 4 Report

The revised manuscript is improved and all comments have been satisfactorily addressed. However, the total number of participants is still inconsistent (268 in text) but the sum of all of groups is 258 (83+79+50+46). Please, revise it again.